# The meaning of home questionnaire revisited: Psychometric analyses among people with Parkinson's disease reveals new dimensions

Nilla Andersson[1,2]*, Maria H. Nilsson[1,2,3], Björn Slaug[1], Frank Oswald[4], Susanne Iwarsson[1]

**1** Department of Health Sciences, Lund University, Lund, Sweden, **2** Memory Clinic, Skåne University Hospital, Malmö, Sweden, **3** Clinical Memory Research Unit, Department of Clinical Sciences Malmö, Lund University, Lund, Sweden, **4** Interdisciplinary Ageing Research (IAW), Goethe University Frankfurt, Frankfurt, Germany

\* nilla.andersson@med.lu.se

**Data Availability Statement:** The data used in this study contains sensitive information about the study participants and they did not provide consent

## Abstract

### Background

Research addressing perceptions of housing in people with Parkinson's disease (PD) is rare, and existing instruments capturing perceived aspects of housing are rarely used. Perceived housing comprises of several domains and is associated with health in general older populations. One such domain is meaning of home, captured by the Meaning of Home Questionnaire (MOH). The aim of this study was to evaluate psychometric properties of the MOH among people with PD.

### Materials and methods

The MOH was administrated to 245 participants with PD (mean age = 69.9 years; mean PD duration = 9.7 years). The instrument consisted of four sub-scales with 28 items, each with 11 response options (strongly disagree = 0; strongly agree = 10). We evaluated data quality, structural validity (factor analysis), construct validity (i.e., testing correlations with relevant constructs according to pre-defined hypotheses), corrected item total correlations, floor and ceiling effects and internal consistency.

### Results

The data quality was high (0–1.2% missing data). The exploratory factor analysis suggested removal of five items and revealed three new factors; "My home is my castle", "My home is my prison" and "My home is my social hub". The 23-item MOH showed statistically significant correlations with life satisfaction, usability and ADL dependence, while not correlated with number of environmental barriers. These findings were largely as hypothesised, thus supporting construct validity (both convergent and discriminant). The corrected item total correlations were >0.3 for all items and the internal consistency was >0.70 for all sub-scales. No floor or ceiling effects were reported except for the sub-scale "My home is my castle" (ceiling effect = 15.6%).

for public data sharing. The current approval by the Regional Ethical Review Board in Lund, Sweden (No. 2012/558) does not include data sharing. A minimal data set could be shared by request from a qualified academic investigator for the sole purpose of replicating the present study, provided the data transfer is in agreement with EU legislation on the general data protection regulation and approval by the Swedish Ethical Review Authority. Contact information: Department of Health Sciences, Lund University Box 157, 221 00 Lund, Sweden DHSdataaccess@med.lu.se Principal investigator: Maria_H.Nilsson@med.lu.se Swedish Ethical Review Authority, Box 2110, 75 002 Uppsala, Sweden. Phone: +46 10 475 08 00.

**Funding:** The HHPD project was funded by the Strategic Research Area in neuroscience (MultiPark) at Lund University (MN), the Swedish Research Council (SI; grant number:2015-02616), Swedish Research Council (SI; 2013-08755-11, K2014-99X-22618-01-6), Neuro (SI), Neuro (MN), Ribbingska Foundation in Lund (SI), Greta and Johan Kock Foundation (MN), Swedish Parkinson Foundation (MN), Norrbacka-Eugenia Foundation (MN) and the Crafoord Foundation (MN). The funders had no role in study design, data collection and analysis, decision to publish, or preparation of the manuscript.

**Competing interests:** The authors have declared that no competing interests exist.

## Conclusions

The 23-item MOH version with three new sub-scales is sufficiently reliable and valid for use in PD populations. This paves the way for further research of meaning of home among people with PD, using the 23-item MOH version.

## Introduction

When using data collection instruments in research and clinical settings, it is essential that they are psychometrically sound. As psychometric properties are sample dependent, it is important to evaluate this when used in new populations [1]. The Meaning of Home Questionnaire (MOH) captures perceived aspects of housing. This instrument has been psychometrically evaluated for research on housing and health along the process of ageing among community-dwelling older people, but not in diagnose-specific populations such as people with Parkinson´s disease (PD).

### Perceived aspects of housing

In order to further the understanding of housing and health dynamics in later life, it is necessary to capture a range of housing aspects–objective as well as perceived [2, 3]. A common approach when studying housing has been to address the physical environment in terms of environmental barriers, accessibility and housing adaptations [4, 5]. Less commonly addressed are perceived aspects of housing [6], even though existing research shows that such aspects are related to independence in daily activities and well-being among older people [2, 7].

Perceived housing is an umbrella term, covering several concepts. Oswald and colleagues suggested a four-domain model of perceived aspects of housing, including housing satisfaction, usability in my home, external housing-related control beliefs and meaning of home [6, 8]. The external sub-scale of the Housing-related Control Beliefs Questionnaire (HCQ) [9] has recently been psychometrically evaluated among people with PD [10]. To continue examining the instruments in the four-domain model psychometrically, this study will address the MOH for people with PD.

The concept meaning of home is a symbolic representation of space, place and personal meaning linked to one´s home [11] and suggests that the home is not just an objective function but also related to individual experiences. With theoretical underpinnings from psychology and environmental gerontology [12–14], the development of MOH [11, 15, 16] was initiated with a data collection including three groups (N = 126; age 61–92 years), one with good health and two with functional limitations (i.e., severe mobility impairment and blindness). Individual face-to-face interviews were administrated at home visits [15, 16] with the following open-ended question: "What makes your house a home?". Four researchers with expertise in psychology and gerontology individually categorised a total of 1804 statements extracted from the interviews based on a qualitative content analysis methodology [17]. Each statement was coded into categories with satisfying interrater reliability (Cohen's Kappa: 0.77–0.83) [16, 18]. The same procedure was administered to reduce the thirteen initial categories into the subsequent five categories of meaning (physical, behavioural, cognitive, emotional, social meaning) resulting in the first version of the 28-item MOH. Later analysis supported treating cognitive and emotional aspects of meaning of home as one category due to content overlaps. The four sub-scale MOH was used in an ENABLE-AGE project sub-study involving community-dwelling very old people from Sweden, Germany and the United Kingdom (n = 1223) [6]. Because

the internal consistency reliability of the social meaning sub-scale was low ($\alpha$ = .44) it was discarded in that study. As to construct validity the remaining sub-scales showed poor to fair significant correlations with other domains of perceived housing (usability in my home; external housing-related control beliefs; housing satisfaction) [6].

Oswald and Wahl argued that perceived housing aspects such as meaning of home, are crucial for higher life satisfaction in old age [12]. Further, a study among community-dwelling very old people suggested that those who have higher scores MOH are more independent in daily life and have a better sense of psychological well-being [2]. Similar results were found in a sample of younger old adults (aged 67–70), where the sub-scales physical, behavioural and social meaning were significantly associated with psychological well-being [19]. Moreover, research using the MOH suggests that meaning of home rating patterns may change over time [12] and should therefore not be considered as temporally consistent. For instance, based on qualitative in-depth interviews Kylén and colleagues [20] pointed out that thoughts on future housing in later life, including issues of meaning of home exists already at the time of retirement.

## Parkinson's disease and meaning of home

PD is a chronic and progressive neurodegenerative disease which is characterized by motor-symptoms [21] and non-motor symptoms [22]. Even at an early stage of the disease, the ADL performance can be negatively affected [23]. People with PD have more difficulties performing more complex activities than controls, for example in housing maintenance and shopping [23]. For people with PD who often have several functional limitations, aspects of the environment may on one hand provide challenges to cope with, on the other hand provide compensatory potential, for example, assistive devices or housing design features. Still, matters related to housing have had scarce attention in PD research. As suggested by Rubinstein, with ageing facets of the housing environment might partially be perceived as merged with the body ("embodiment") [24], such as hearing aids, a cane or favourite places (e.g., a sofa). The idea of blurry boarders between the body and the environment is particularly important in old age when the home symbolises high amounts of personal values and meaning. While true for housing in later life in general [24], embodiment may be especially important for people with a progressive neurodegenerative disease such as PD as the housing environment may enable activities that otherwise would be difficult or even impossible to perform. When it comes to research on housing and PD, results show that people with PD have a five-fold higher risk of moving to assisted living facilities than the general population and they often move there at an earlier age [25], but little is known about housing and health dynamics in this sub-group of the ageing population.

The MOH has been used in PD studies attempting to capture meaning of home, but without prior psychometric evaluation for this specific population. In a first study, Nilsson and colleagues [26] found that very old people with self-reported PD (N = 20) perceived less physical and behavioural meaning of home compared to matched controls (N = 60), but no differences were found in terms of cognitive/emotional and social meaning. A second study indicated that among people with verified PD (N = 231) those who had more functional limitations perceived their housing as less meaningful regarding behavioural aspects [27].

In order to identify whether the MOH validly and reliably captures meaning of home among people with PD, this study aimed to evaluate the following psychometric properties of the instrument: data quality, structural validity, construct validity (convergent and discriminant), corrected item total correlations, floor and ceiling effects and internal consistency reliability.

## Materials and methods

### Participants and recruitment

We used baseline data from the longitudinal cohort study "Home and Health in People Ageing with Parkinson´s Disease" (HHPD), for details see the study protocol [28]. A sample of 653 potential participants met the inclusion criterion of being diagnosed with PD (ICD 10-code G20.9) since at least one year. They were recruited from three hospitals in Skåne County, Sweden. Out of these, 58 lived outside the study district and were excluded. Further exclusion criteria were having difficulties understanding/speaking Swedish (n = 10); severe cognitive problems (n = 91) or other reasons (n = 57, e.g., recent stroke) that made them unable to give informed consent or take part in the majority of the data collection. Both the inclusion- and exclusion criteria were evaluated by a specialized PD-nurse and by screening of medical records. The remaining potential participants (n = 437) were invited to participate. However, 22 were unreachable and two had their PD diagnosis changed. Moreover, 157 of the 413 participants declined to participate and one had extensive missing data (n = 255). For the present study, eight persons were excluded due to limited answers on core variables and another two because they lived in residential care units. Thus, the final study sample was N = 245 participants (60.8% men; mean age = 69.9 years; min-max = 45–93 years) with a median PD duration of 8 years (q1-q3 = 5–13) (Table 1). Selected descriptive variables were used to characterize the sample, such as cognitive functioning [29], depressive symptoms [30], PD specific motor symptoms [31] and housing variables (see Table 1).

We followed the principles of the Helsinki Declaration, and each participant provided written informed consent. The HHPD project was approved by the Regional Ethical Review Board in Lund (No. 2012/558).

### Data collection

Two project administrators who underwent project-specific training collected the data through a postal survey and at a subsequent home visit with each participant. The postal survey was conducted about 10 days in advance of the home visit, when the MOH was administered.

### Instruments

**Meaning of home (MOH) questionnaire.**  The MOH is a questionnaire administered by specifically trained project assistants from different professional backgrounds. The instrument has four sub-scales with 28 items: physical (7 items), behavioural (6 items), cognitive/emotional (10 items) and social (5 items) meaning. The items are presented with response options, on an 11-point scale, ranging from strongly disagree (0) to strongly agree (10) and marked with endpoints. Higher scores indicate perceiving more meaning. Three items are negatively phrased (items no. 11, 16, 21) and eight items are reversed (items no. 6, 9, 11, 15, 19, 21, 25, 27). The latter are inverted when processing the data [11, 12]. Psychometric properties of the MOH questionnaire for use in the general population of older people have been reported earlier [6]. The questionnaire [32] is available in English, Swedish, and German versions for research application (on request only, to author F.O.).

In the present study, we calculated a sum score (adding up the scores from the items) and a mean score (dividing the sum score with the number of items) for each sub-scale. If answers of one or more items were missing, a sum or mean score was not calculated. The MOH questionnaire was assessed in face-to-face sessions during home visits in this study. We used the Swedish version of the MOH, which emanated from an iterative translation process conducted by a multi-lingual research team [6]. It should be noted that the social sub-scale was used albeit the low internal consistency values reported earlier [6].

**Table 1. Sample characteristics, N = 245.**

| Variable | *n* (%) unless otherwise stated | Missing, *n* |
|---|---|---|
| **Participant characteristics** | | |
| Sex, women/men | 96 (39.2)/149(60.8) | - |
| Age, mean (SD) | 69.7 (9.0) | - |
| PD duration (years), median (q1-q3) | 8 (5–13) | 1 |
| Disease severity (HY during on phase) | | - |
| *HY I* | 50 (20.4) | |
| *HY II* | 73 (29.8) | |
| *HY III* | 62 (25.3) | |
| *HY IV* | 54 (22) | |
| *HY V* | 6 (2.5) | |
| Motor symptoms (UPDRS III), median (q1-q3) | 29 (22–39) | 4 |
| Global cognitive function (MOCA), median (q1-q3) | 26 (23–28) | 5 |
| ADL (ADL Staircase), median (q1-q3) | 4 (0–8) | - |
| Depressive symptoms (GDS 15), median (q1-q3) | 3 (1–4) | 5 |
| Higher education (university), yes/no | 83 (33.9)/162 (66.1) | - |
| Life satisfaction (item 1, Lisat -11), median (q1-q3) | 5 (4–5) | - |
| **Housing characteristics** | | |
| Type of housing | | - |
| *Apartment* | 109 (44.5) | |
| *Housing* | 131 (53.5) | |
| *Other* | 5 (2) | |
| Residential location | | - |
| *Rural* | 79 (32.2) | |
| *Semi-Urban* | 65 (26.5) | |
| *Urban* | 101 (41.3) | |
| Tenure of housing | | - |
| *Privately owned/rental* | 184 (75.1)/61 (24.9) | |
| Accessibility problems (HE), median (q1-q3) | 185 (95–272) | 1 |
| Housing adaptation, yes/no | 80 (32.7)/165 (67.3) | - |
| Years in present dwelling, median (q1-q3) | 17 (5–35) | - |

HY = Hoehn & Yahr, possible scores 1–5 (higher = worse disease severity). UPDRS = Unified Parkinson's disease rating scale, part III, possible scores: 0–108 (higher = more motor symptoms). MOCA = Montreal Cognitive Assessment, possible scores 0–30 (higher = better cognitive function). ADL = Activities of Daily Living Staircase, possible total sum score 0–27 (higher = more dependent). GDS = Geriatric Depression Scale, possible total sum score 0–15 (higher scores = more depressive symptoms). Lisat -11 = Life Satisfaction Questionnaire, item 1, possible scores: 1–6, (higher = greater life satisfaction). HE = Housing Enabler, possible scores 0–1844 (higher = greater accessibility problems).

## Construct validity hypotheses and variables used

To test convergent and discriminant aspects of construct validity predefined hypotheses were formulated based on previous research [2, 6, 12] and clinical reasoning and assessed with related concepts.

Targeting convergent validity we defined four hypotheses. Firstly, we expected more perceived meaning (i.e., higher MOH scores) to correlate with higher life satisfaction [12], assessed with item 1 of Lisat -11: "Life as a whole is. . ..." [33]. The scale ranges from very dissatisfying (1) to very satisfying (6); higher scores indicate more life satisfaction. Secondly, we

hypothesized that more perceived meaning would correlate with positively with perceived usability [6], assessed with the Usability in My Home questionnaire (UIMH) [34, 35]. This instrument addresses to what extent the environment supports the performance of activities at home. We used two sub-scales of the UIMH; "activity aspects" (4 items) and "physical environment" (6 items). Each item has 6 response options (from 0 = not at all to 5 = fully agree); higher scores reflect more usability. Thirdly we hypothesised, higher scores on MOH (more perceived meaning) for those who had lived longer in their current dwelling (i.e., living duration in number of years). This was based on reasoning by Oswald and colleagues about perceived meaning of home from a lifespan perspective [12]. To capture time in the same dwelling, we used the continuous variable "number of years in the same dwelling" from a question in the data collection. Fourthly, we expected that more perceived meaning (i.e. higher MOH scores) would correlate with less ADL dependence [2]. ADL dependence was assessed with the ADL Staircase [36, 37], consisting of nine items (personal ADL: feeding, transfer, toileting, dressing and bathing; instrumental ADL: cooking, transportation, shopping and cleaning). Each item was assessed on a three-graded scale: independent/partly dependent/dependent. If the response option independent was used, the participants were asked to specify if the activity was performed with or without difficulties [38]. The items of the ADL Staircase were summed up to a total sum score (range 0–27); higher scores reflect more dependency.

Addressing discriminant validity to differentiate the meaning of home from other concepts [39], in the fifth hypothesis we expected higher MOH scores to not correlate significantly with more environmental barriers in the housing environment. This objective aspect of housing was assessed with the Housing Enabler (HE) instrument [40] using the dichotomous assessments (present/not present) of 161 physical environmental barriers (higher number = more barriers).

## Data analysis

Participant characteristics were described using median, quartiles and frequencies. The choice of statistical tests to evaluate the psychometric properties of the MOH instrument was based on the COSMIN checklist [41].

**Data quality.** Data quality was calculated for each sub-scale and presented as the percentage of missing data for items and sub-scale sum scores [1].

**Structural validity.** Exploratory factor analysis was used to evaluate structural validity. Because the data was ordinal and not normally distributed, we used a principal axis factoring with an oblimin rotation with Kaiser Normalization. To determine how many factors to include in the model we did a visual examination of the scree plot and used the following criteria: no factor with less than three items, few cross-loadings (i.e., high loadings on more than one item) and items with all factor loadings <0.33 [42]. In case the factor analysis would indicate that the instrument comprised items not fulfilling these criteria, we were prepared to take further steps to optimize structural validity by conceptual considerations and knowledge from an earlier study [11] in relation to the present study. The factor solution that emerged was first individually assessed by each of the five authors (representing different professions and expertise), in terms of coherence and labelling of the factors. The individual assessments were then considered in joint consensus discussions until agreements was achieved on coherence and how to label of the factors.

**Construct validity (Convergent and discriminant validity).** Convergent and discriminant validity was analysed using Spearman´s rank correlation coefficient ($r_s$) [43].

**Corrected item total correlation.** To evaluate the correctness of summing up the items to a total score for each sub-scale, we used >0.3 as a reference value for the corrected item-total

correlations [1]. Values >0.4 indicate that the sub-scale is measuring the same underlying construct [1].

**Floor and ceiling effects.** Floor and ceiling effects (i.e., the percentage respondents who received the minimum and maximum possible score, respectively) were calculated for the sum sub-scale score [44] using 15–20% as acceptable reference value [1]. On item level, floor and ceiling effects were analysed through the distribution of response alternatives. The only published reference value found was a 75% limit, which was used for interpretation [45], albeit it may be somewhat high.

**Cronbach´s alpha.** Cronbach´s alpha was used to evaluate internal consistency reliability, with values >0.70 considered acceptable [1]. The standard error of measurement (SEM) was calculated with the formula $SD_{baseline} X \sqrt{1 - reliability}$, and complemented with a 95% confidence interval [1].

For all analyses, p values <0.05 were used to define statistical significance. For computations, we used the IBM SPSS statistics 25 software (IBM Corporation, Armonk, NY, USA).

## Results

The median (q1-q3) sum scores of the 28-item MOH sub-scales were as follows: physical meaning 53 (49–58); behavioural meaning 46 (40–54); cognitive/emotional meaning 80 (72–88) and social meaning 43 (37–48).

### Data quality and structural validity

For the 28-item MOH, the overall data quality was high, with few missing item responses: 0.4% for 5 items, 0.8% for 14 items and 1.2% for one item. Eight items had no missing values.

After exploratory testing with different numbers of factors, using the aforementioned criteria the result was a clear three-factor solution explaining 33% of the variance, see Table 2. Items 1 ("Living in a place which is well-designed and geared to my needs"), 4 ("Feeling safe") and 22 ("Thinking about that living here will be like in the future") had low corrected item correlations values and low factor loadings (<0.33). Further, item 2 ("Managing things without the help of others") had a low factor loading and item 17 ("Thinking about the past") loaded on all three factors just below the cross loading value. Accordingly, these five items were excluded. A second factor analysis was made with the remaining items (Table 2), which resulted in a similar three-factor solution. The second factor analysis was considered as conceptually meaningful by the authors, explaining 36% of the variance.

Results presented henceforth are based on the new 23-item version of the MOH. The items of the three new factors were used as summed sub-scales based on the result of the factor analysis. Based on an iterative discussion among the authors, the sub-scales were labelled: "My home is my castle" (8 items, eligible scores 0–80), "My home is my prison" (8 items, eligible scores 0–80) and "My home is my social hub" (7 items, eligible scores 0–70).

With regard to item movements from the 28- to the 23-item MOH, the new sub-scale "My home is my prison" consists of items from all four of the original sub-scales, but mostly from the physical sub-scale. There were no items moving from the physical or social sub-scale to the sub-scale "My home is my castle", instead the items came from the behavioural and cognitive/emotional sub-scales. The sub-scale "My home is my social hub" got most items from the physical and social sub-scales and just one item from the cognitive/emotional sub-scale (see Fig 1).

As to the 23-item version of the MOH, the median (q1-q3) scores for the new sub-scales were "My home is my castle" 68 (59–76), "My home is my prison" 68 (59–75), "My home is my social hub" 58 (50–65). There were few missing item responses; 5 items had no missing values, 5 had 0.4%, 12 had 0.8% and 1.2% for one item (see Table 3).

Table 2. Exploratory factor analysis with oblimin rotated factor loadings of the 23 items version MOH.

| Factor 1 –My home is my castle (8 items) | Factor 1 | Factor 2 | Factor 3 |
|---|---|---|---|
| 18. Enjoying my privacy and being undisturbed | 0.67 | 0.06 | -0.09 |
| 26. Being able to do what I please | 0.65 | 0.12 | -0.02 |
| 16. Not having to accommodate anyone´s wishes but my one | 0.64 | -0.14 | 0.06 |
| 13. Being able to change or rearrange things as I please | 0.62 | -0.09 | 0.05 |
| 14. Being able to relax | 0.59 | 0.31 | -0.01 |
| 23. Feeling comfortable and cosy | 0.37 | 0.30 | 0.19 |
| 10. Knowing my home like the back of my hand | 0.34 | 0.02 | 0.30 |
| 8. Doing everyday tasks | 0.33 | 0.13 | 0.25 |
| **Factor 2 –My homes is my prison (8 items)** | Factor 1 | Factor 2 | Factor 3 |
| 21. No longer being able to keep up with the demands of my home | 0.07 | 0.64 | -0.13 |
| 27. Feeling lonely | 0.01 | 0.59 | 0.06 |
| 9. Being bored | 0.01 | 0.54 | -0.10 |
| 19. Being excluded from social and community life | 0.08 | 0.53 | 0.10 |
| 15. Feeling that home has become a burden | 0.03 | 0.49 | 0.04 |
| 25. Being confined to rooms inside the house | -0.09 | 0.47 | 0.01 |
| 11.Living in a place where I can get no support or help from others | -0.02 | 0.43 | 0.08 |
| 6. Having to live in poor housing conditions | 0.01 | 0.39 | 0.16 |
| **Factor 3 –My Home is my social hub (7 items)** | Factor 1 | Factor 2 | Factor 3 |
| 5. Meeting family, friends, and acquaintances | -0.22 | 0.11 | 0.73 |
| 28. Having a good relationship with the neighbours | -0.06 | 0.08 | 0.56 |
| 24. Being able to receive visitors | 0.16 | 0.09 | 0.53 |
| 3. Being familiar with my immediate surroundings | 0.21 | 0.01 | 0.51 |
| 7. Having a nice view | 0.11 | -0.14 | 0.47 |
| 12. Living in a place that is comfortable and tastefully furnished | 0.16 | 0.06 | 0.45 |
| 20. Having a base from which I can pursue activities | 0.33 | 0.14 | 0.33 |

*Five items (nos. 1, 2, 4, 17, 22) were removed from the original MOH instrument. Three due to factor loadings <0.33 (nos. 1, 2, 4), one (no. 22) due to low corrected item total values and one due to cross loadings (item 17).

## Construct validity (Convergent and discriminant)

The results regarding convergent validity showed, as expected from the pre-defined hypotheses, that higher MOH scores were poorly to fairly correlated with higher life satisfactions, higher usability and lower ADL dependence ($p<0.05$). "My home is my social hub" had a poor ($r_s = 0.21$) but statistically significant correlation with time lived in the same dwelling, while the sub-scales "My home is my castle" and "My home is my prison" were not. As for the discriminant validity, the very low non-significant correlations between higher MOH scores and number of environmental barriers ($r^s = -0.069$ to-0.096) were as expected. For details, see Table 4.

## Corrected item total correlations

For the 23-item MOH version, all items had corrected item total correlation values >0.3 and only two items had values <0.4 (see Table 3).

## Floor and ceiling effects

No floor or ceiling effects were found on item level. On sub-scale level we found a ceiling effect (15.6%) for the sub-scale "My home is my castle" (see Table 3). The other two sub-scales had no floor or ceiling effects.

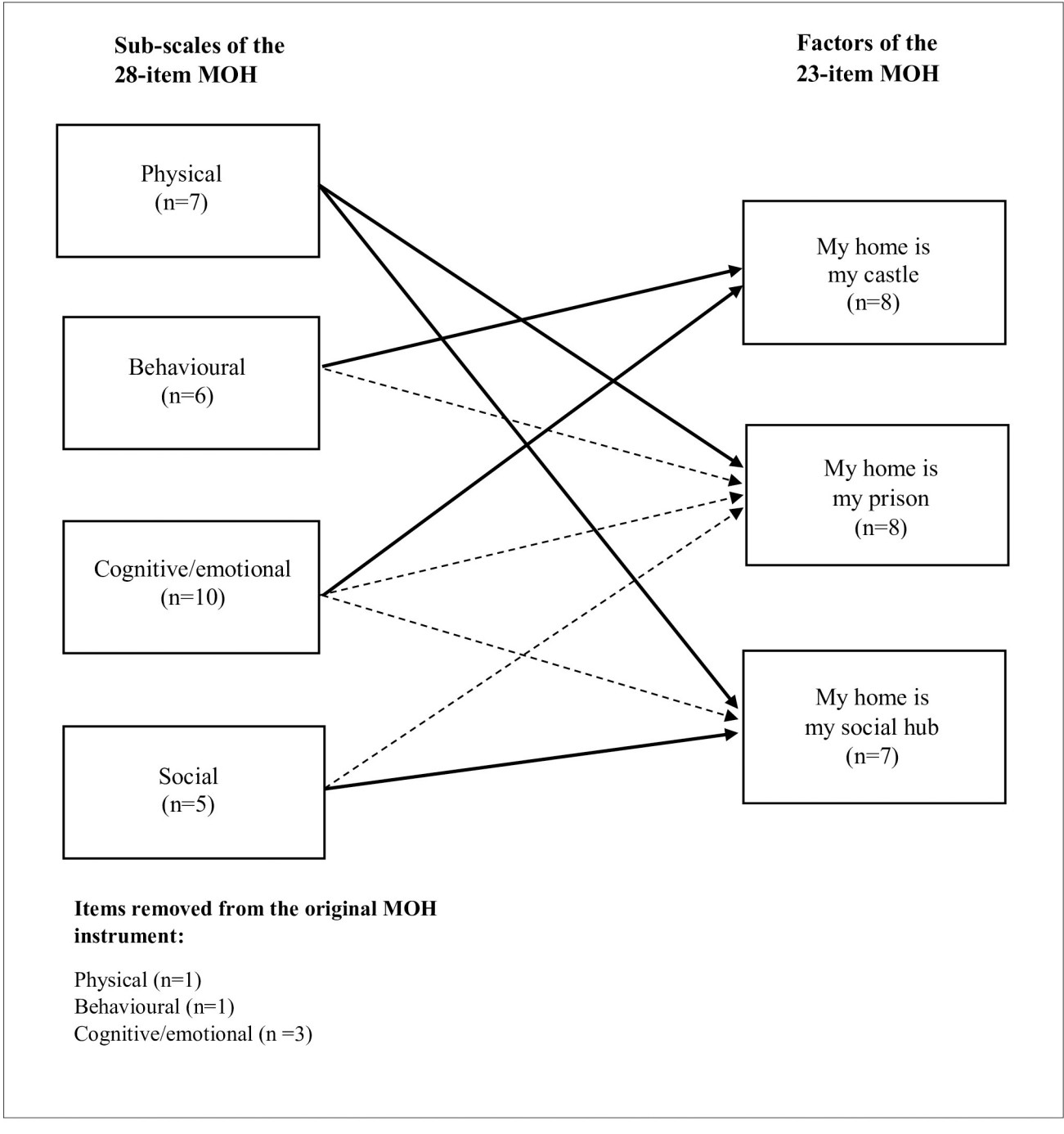

= 1-2 items; ⟶ = 3-4 items.

**Fig 1. Items movement from the sub-scales of the 28-item version of MOH, to the new factor solution.**

### Internal consistency reliability

For the 23-item MOH, all three sub-scales had Cronbach's alpha values >0.70. The SEM score range was 5–6 for the sum score (see Table 3).

**Table 3. Results of psychometric analysis for the 23-item MOH instrument after factor analysis, N = 245.**

| Being at home means for me…. | Median (q1-q3) | Missing (n) | Floor effects, n (%) | Ceiling effects, n (%) | Corrected item-total correlation |
|---|---|---|---|---|---|
| **Sub-scale 1: My home is my castle** | | | **Response options** | | |
| | | | "Strongly disagree" | "Strongly agree" | |
| 8. Doing everyday tasks | 8 (7–10) | - | 1 (0.4) | 82 (33.5) | 0.46 |
| 10. Knowing my home like the back of my hand | 9 (8–10) | 1 | 2 (0.8) | 118 (48.4) | 0.43 |
| 13. Being able to change or rearrange things as I please | 9 (7–10) | 1 | 3 (1.2) | 92 (37.7) | 0.52 |
| 14. Being able to relax | 10 (9–10) | 2 | 3 (1.2) | 144 (59.3) | 0.62 |
| 16. Not having to accommodate anyone´s wishes but my one | 8 (5–10) | 2 | 12 (4.9) | 85 (35) | 0.54 |
| 18. Enjoying my privacy and being undisturbed | 9 (7–10) | 2 | 4 (1.6) | 105 (43.2) | 0.54 |
| 23. Feeling comfortable and cosy | 10 (8–10) | 2 | 1 (0.4) | 131 (53.9) | 0.50 |
| 26. Being able to do what I please | 10 (8–10) | 2 | 2 (0.8) | 123 (50.6) | 0.61 |
| **Sub-scale** | | | 0 (0.0) | 38 (15.6) | |
| **Sum score, median (q1-q3)** | 68 (59–76) | | | | |
| **SEM (CI 95%)** | 5 (-5; 15) | | | | |
| **Mean score, mean (SD)** | 8.3 (1.4) | | | | |
| **SEM (CI 95%)** | 0.6 (-0.6; 1.8) | | | **Cronbach´s alpha 0.80** | |
| **Sub-scale 2: My home is my prison** | | | | | |
| 6. Having to live in poor housing conditions | 10 (9–10) | - | 1 (0.4) | 167 (68.2) | 0.41 |
| 9. Being bored | 10 (8–10) | - | 1 (0.4) | 129 (52.7) | 0.41 |
| 11. Living in a place where I can get no support or help from others | 10 (8–10) | 1 | 3 (1.2) | 123 (50.4) | 0.38 |
| 15. Feeling that the home has become a burden | 10 (8–10) | 2 | 2 (0.8) | 133 (54.7) | 0.44 |
| 19. Being excluded from social and community life | 10 (8–10) | 2 | 3 (1.2) | 142 (58.4) | 0.53 |
| 21. No longer being able to keep up with the demands of my home | 8 (5–10) | 2 | 5 (2.1) | 84 (34.6) | 0.52 |
| 25. Being confined to rooms inside the house | 8 (4–10) | 2 | 21 (8.6) | 78 (32.1) | 0.39 |
| 27. Feeling lonely | 9 (7–10) | 2 | 2 (0.8) | 114 (46.9) | 0.51 |
| **Sub-scale** | | | 0 (0) | 30 (12.3) | |
| **Sum score, median (q1-q3)** | 68 (59–75) | | | | |
| **SEM (CI95%)** | 6 (-6; 17) | | | | |
| **Mean score, mean (SD)** | 8.2 (1.4) | | | | |
| **SEM (CI95%)** | 0.7 (-0.7; 2.2) | | | **Cronbach´s alpha 0.74** | |
| **Sub-scale 3: My home is my social hub** | | | | | |
| 3. Being familiar with my immediate surroundings | 9 (8–10) | 1 | 2 (0.8) | 104 (42.6) | 0.55 |
| 5. Meeting family, friends and acquaintances | 9 (8–10) | - | 5 (2) | 113 (46.1) | 0.53 |
| 7. Having a nice view | 8 (5–10) | - | 9 (3.7) | 69 (28.2) | 0.40 |
| 12. Living in a place that is comfortable and tastefully furnished | 9 (7–10) | 1 | 3 (1.2) | 99 (40.6) | 0.48 |
| 20. Having a base from which I can pursue activities | 9 (7–10) | 3 | 3 (1.2) | 99 (40.9) | 0.43 |
| 24. Being able to receive visitors | 10 (8–10) | 2 | 1 (0.4) | 138 (56.8) | 0.55 |
| 28. Having a good relationship with the neighbours | 8 (6–10) | 2 | 7 (2.9) | 76 (31.1) | 0.47 |
| **Sub-scale** | | | 0 (0) | 20 (8.3) | |
| **Sum score, median (q1-q3)** | 58 (50–65) | | | | |
| **SEM (CI 95%)** | 5 (-5; 15) | | | | |
| **Mean score, mean (SD)** | 8.1 (1.5) | | | | |

*(Continued)*

**Table 3.** (Continued)

| Being at home means for me. . .. | Median (q1-q3) | Missing (n) | Floor effects, n (%) | Ceiling effects, n (%) | Corrected item-total correlation |
|---|---|---|---|---|---|
| **SEM (CI 95%)** | 0.7 (-0.7; 2.1) | | | Cronbach´s alpha 0.76 | |

MOH = Meaning of Home Questionnaire. Sub-scale "My home is my castle", sum score = adding up the scores from the items of the sub-scale. Possible sum scores: 0–80 (higher = perceiving more meaning). Mean score = adding the scores from the items and dividing it with the number of items of the sub-scale. Possible mean scores: 0–10 (higher = perceiving more meaning). Sub-scale "My home is my prison" = the scores are inverted in all items (higher scores = perceiving more meaning). Possible sum scores: 0–80 (higher = perceiving more meaning). Possible mean scores: 0–10 (higher = perceiving more meaning). "My home is my social hub" = possible sum scores: 0–70 (higher = perceiving more meaning). Possible mean scores: 0–10 (higher = perceiving more meaning).

## Discussion

This is the first study evaluating psychometric properties of the MOH among people with PD. Building on existing conceptual frameworks within psychology and environmental gerontology [12–14], this study contributes to the methodological development for research targeting

**Table 4. Results of correlations between the 23-item version of the MOH instrument and life satisfaction, usability in the home, how long the participants have lived in their dwelling ADL dependence and number of environmental barriers.**

| Hypothesis | Sub-scales | Correlation coefficient ($r_s$), p-values | Hypothesis confirmed |
|---|---|---|---|
| *Convergent validity* Higher MOH scores expected to correlate significantly with higher life satisfaction (item 1, Lisat-11) | MOH 1 | 0.32, <0.001 | Yes |
| | MOH 2 | 0.43, <0.001 | Yes |
| | MOH 3 | 0.31, <0.001 | Yes |
| Higher MOH scores expected to correlate significantly with low to moderate strength with higher scores of Usability in My Home (UIMH). | MOH 1— UIMH 1 | 0.28, <0.001 | Yes |
| | MOH 1— UIMH 2 | 0.45, <0.001 | Yes |
| | MOH 2— UIMH 1 | 0.26, <0.001 | Yes |
| | MOH 2— UIMH 2 | 0.35, <0.001 | Yes |
| | MOH 3— UIHM 1 | 0.31, <0.001 | Yes |
| | MOH 3— UIMH 2 | 0.45, <0.001 | Yes |
| Higher MOH scores expected to correlate significantly for those who have lived longer in their dwelling (number of years) | MOH 1 | 0.05, = 0.413 | No |
| | MOH 2 | -0.07, = 0.258 | No |
| | MOH 3 | 0.21, <0.001 | Yes |
| Higher MOH scores expected to correlate significantly with lower ADL dependence. | MOH 1 | -0.18, <0.004 | Yes |
| | MOH 2 | -0.33, <0.001 | Yes |
| | MOH 3 | -0.19, 0.003 | Yes |
| *Discriminant validity* Higher MOH scores expected to not correlate significantly with number of environmental barriers in the dwelling (HE). | MOH 1 | -0.07, = 0.257 | Yes |
| | MOH 2 | -0.07, = 0.282 | Yes |
| | MOH 3 | -0.10, = 0.135 | Yes |

MOH = Meaning of Home Questionnaire, 23-item version with 3 sub-scales. Lisat -11 = Life satisfaction Questionnaire, item 1, possible scores: 1–6, (higher = greater life satisfaction). MOH 1 = MOH Sub-scale: "My home is my castle", possible sum scores: 0–80 (higher = perceiving more meaning). MOH 2 = MOH Sub-scale: "My home is my prison", possible sum scores 0–80 (higher = perceiving more meaning). MOH 3 = MOH Sub-scale: "My home is my social hub", possible sum scores 0–70 (higher = perceiving more meaning). UIMH = Usability in My Home. UIMH 1 = UIMH Sub-scale: activity aspects, possible scores 4–20 (higher = more usable). UIMH 2 = UIMH Sub-scale: physical aspects, possible scores 6–30 (higher = more usable). ADL = Activities of Daily Living, measured with ADL Staircase. Possible total sum score 0–27 (higher = more dependent). Number of environmental barriers, possible score 0–161 (higher = more environmental barriers).

housing and health dynamics, specifically regarding aspects of perceived housing. The exploratory factor analysis of the 28-item MOH suggested removal of five items and revealed three new meaningful dimensions, which were labelled "My home is my castle", "My home is my prison" and "My home is my social hub". The results support the construct validity of the suggested 23-item version of MOH and are largely in line with the pre-defined hypotheses. As to the homogeneity analysis, the results support items being summed up to a sum score for each of the three new sub-scales. The data quality was generally high, with no floor or ceiling effects for two of the sub-scales and only a border-line ceiling effect for the sub-scale "My home is my castle". Furthermore, the internal consistency reliability values were acceptable. Taken together, the results indicate that the 23-item MOH can be reliable and validly used among people with PD.

Compared to the 28-item MOH [11, 12], the three sub-scale structure of the instrument is entirely different form that of the former version. The original sub-scales were constructed according to pre-defined theoretical concepts while the current new factor solution is based on statistical analysis of empirical data collected with a sample of the population at target. As to the removal and restructuring of items, it is intriguing that the items of the cognitive/emotional sub-scale were distributed on all three sub-scales in the 23-item version. This might reflect that the cognitive and emotional aspects of meaning of home are important within different everyday life context, at least for the current sample. That is the cognitive and emotional aspects did not form a distinct factor in the analysis itself, as was argued before [18]. Further, the sub-scale "My home is my castle" consists only of items from the original behavioural and cognitive/emotional sub-scales of the 28-item MOH, potentially indicating that physical and social aspects are not as important for this sub-scale. A qualitative study among people with PD [46], suggests that cognitive impairments and psychological changes affects the way they manage their everyday activities. This result mirrors the content of the items in the sub-scale "My home is my castle". In contrast, items from all sub-scales of the 28-item MOH contributed to the new sub-scale "My home is my prison", but the largest number of items came from the former physical sub-scale [18]. Results showing that people with PD have more accessibility problems in their housing environments than controls [26] speaks to the relevance of studying meaning of home in this population. That is, a physical housing environment that does not match the person's functional limitations could lead to negative perceptions that are possible to capture with the sub-scale "My home is my prison". As for the sub-scale "My home is my social hub", interestingly, three items came from the physical sub-scale and the same number of items came from the social sub-scale of the 28-item MOH (see Fig 1). One of the items from the physical sub-scale of the 28-item MOH, which moved to the sub-scale "My home is my social hub" (no. 20; Having a base from where I can pursue activities) addresses physical activities. A previous study among people with PD described social embarrassment as related to physical limitations such as involuntary movement, less facial expressions and tremor [46] and other studies have shown the importance of walking in relation to social contexts [47, 48]. These results suggest that physical aspects and social context are related, which is in line with our findings of the sub-scale "My home is my social hub". An aspect supporting the use of the 23-item MOH among people with PD is the homogeneity of the sub-scales. The corrected item total values of the 23-item MOH are stronger (all corrected item total values >0.30), compared to the 28-item version where nine items had corrected item total values <0.30 [1]. In summary, the three new suggested factors in the 23-item version are interpreted as valid sub-scales of the MOH used among people with PD.

Moreover, the results of the factor analysis suggested a removal of five items for various reasons. Three of these items (nos. 1, 4, 22) had low factor loadings (<0.33) and corrected item correlations (<0.3) whereas one item (no. 2) had a very low corrected item total value.

Furthermore, item no. 17 ("Thinking about the past") had similar loadings on all three factors, making it difficult to interpret which factor that the item truly belonged to [42]. Moreover, an earlier study showed that item 17 had conceptual difficulties [11]. For both these reasons item 17 was therefore excluded in the 23-item MOH. The removal of five items is positive from a respondent burden perspective, especially for vulnerable participants groups [45]. People with PD often have non-motor symptoms such as fatigue, pain or depression [22] and a shorter instrument might make it possible for more and frailer participants to participate in research. Important to highlight, however, is that two items with cross loadings were retained in the 23-item version (i.e. no. 10; "Knowing my home like the back of my hand", and no. 23; "Feeling comfortable and cosy"). Both these items had cross-loadings on two of the sub-scales, but each was possible to place in one of the sub-scales in a conceptually meaningful way. These two items had a strong impact in the development phase of the MOH, and were considered as "signature items", i.e., reflecting core opinions by many participants throughout the instrument development process. Nevertheless, factor analysis is a crucial foundation when evaluating psychometric properties of an instrument.

Regarding some underlying theoretical aspects of meaning of home, Rubinstein's reasoning about the close-to-body experience [24] might be of particular importance for people with neurodegenerative limitations. Relating this to the content of the new sub-scale "My home is my castle", strong agreement with the item "Knowing my home like the back of my hand" could be interpreted as related to close-to-body features of the environment strengthening the perceived meaning of home. Another concept mentioned by Rubinstein is environmental centralisation, which could be linked to the sub-scale "My home is my prison" with items such as "Being confined with the rooms (and things) inside my home" and "No longer being able to keep up with the demands of my home". That is, with increasing functional limitations a person uses a smaller part of his/her dwelling, which affects the perception of meaning of home in a restrictive way [24]. An example of environmental centralisation is an older adult who spends most of the day in bed or a comfortable chair, surrounded with newspapers, television, a lamp and telephone. This might be particularly relevant for populations such as people with PD, with their explicit challenges related to body functions and activity limitations as the disease progresses.

Regarding the validity evaluation of the 23-item MOH, most correlations with the variables chosen for these analyses supported the construct validity of this version of the instrument. The results showing that the strength of the correlations measuring construct validity is low [39] indicate that meaning of home is distinctly different from usability of the home, ADL dependence and life satisfaction, although related (see Table 4). Accordingly, these results are in line with the pre-defined hypotheses. Comparing with earlier studies in general populations samples of very old people, the correlations in the present study are on approximately similar levels [4, 6]. Reflecting further on, for example, the relation of meaning of home with the well-established and clinically relevant indicators for ADL dependence, meaning of home is based on perceptions of meaningfulness related to the home [11, 12], emerging form psychological theory. In contrast, ADL is a largely physical aspect of performance in everyday activities [36, 37]. Thus, the two concepts have fundamentally different origins, although overlapping in real life situations, and–as demonstrated by previous as well as present results—measuring them requires different instruments.

However, one of the hypotheses regarding convergent validity was only partially confirmed. That is, based on previous theoretical findings [2, 6, 12] we hypothesised that longer time lived in the same dwelling would be associated with higher MOH. This was indeed true for the sub-scale "My home is my social hub", but not for "My home is my castle" and "My home is my prison". To explain this observation one might assume Rowles and Watkin´s "Life Course

Model of Environmental Experience", suggesting that older people frequently report to feel at home quite quickly after moving and preserving high levels of meaning of home without large variation over time [49]. Moreover, Cramm and Neiboer [50] reported that social cohesion and belonging to the neighbourhood are factors that relate to and can predict social well-being among community-dwelling older people. They suggested that when people have lived in a neighbourhood for a long time, it could become a critical support system for health. This kind of reasoning might explain the significant correlation between the sub-scale "My home is my social hub" and time lived in the dwelling. As to the fact that our hypothesis was not supported regarding two of the MOH sub-scales, it seems as other factors than time lived in the dwelling are related to perceiving one's home as a prison or castle, respectively. One example of a redefined hypothesis is that for people living and ageing with a chronic, progressive disease, functional decline is related to perceiving the home as a prison. Obviously, more research is warranted to shed further light on those associations.

As to discriminant validity, the very low and non-significant correlations between MOH sub-scale scores and number of environmental barriers demonstrate that even if the variables represent aspects of housing, self-perceived meaning of home is distinctly different from an objectively measurable variable capturing physical housing features.

When comparing our results with other studies, it should be noted that different versions of the MOH were used (i.e., with different sub-scales and number of items) [2, 6, 8]. Furthermore, because only a small amount of research has addressed meaning of home and related concepts, the theoretical reasoning and findings that we based our hypotheses on are few and somewhat inconsistent. While existing literature seems to presuppose that higher amounts of MOH are positive [2, 6, 12], taking a critical stance this should not be taken for granted. That is, what if high ratings of MOH would actually make them more confined to their house per se and/or prevent people from making proactive decisions to change their housing situation in time before being confined to the home? Relocation has previously been addressed by researchers in relation to attachment or meaning of home. For example, Rowles and Watkins [49] argued that people that have established a meaningful place (home), tends to generate a residential inertia and want to age in place. That is, there is a lower willingness to relocate if one perceives more meaning of home. Building on that, Wahl and Oswald [13] stated that older people perceiving more meaning in relation to the housing could become "prisoners of space" in a no-longer-fitting housing environment. Relocation in very old age has been further described as a complex and ambivalent process [51], which is also true for younger older people [20]. Such dynamics might explain some of the somewhat mixed results of the present study, not the least in particularly challenged groups, such as people ageing with PD. Still, our results support the content validity of the 23-item version of MOH.

Meaning of home is a complex concept [11, 12, 16] to capture in its entirety in one instrument, yet it is not until a validated instrument exists that research can benefit from it. As the 23-item version of MOH seems to be reliable and valid for use among people with PD, one might consider whether it would be valid also for other populations. Could the 23-item MOH be used for people with similar functional limitations, such as other chronic diseases, or are the results of the present study PD-specific? This is an important matter to investigate as psychometric properties often are sample dependent [1]. Furthermore, only a limited number of studies have addressed perceived aspects of home, including the MOH, among people with PD [26, 27]. Our revisiting of the MOH by psychometric analyses and evaluation enables forthcoming research, to understand how perceived aspects of home are associated with health among people with PD. This kind of research delivers new knowledge about housing and health dynamics in a vulnerable sub-group of the ageing population, where people are living for 15–20 years with a progressive disease. As such research is new to the field of PD,

immediate clinical utility is not the target. Rather, knowledge emanating from research based on notions of environmental gerontology [3, 14] has implications on the societal level and the potential to deliver evidence to inform housing interventions and future housing policies accommodating the needs of vulnerable sub-groups.

## Strengths and limitations

Psychometrically sound instruments are important both in research and practice [52]. This study is the first to investigate psychometric properties of the MOH among people with PD. However, to make the psychometric evaluation more complete it should include test-retest reliability as well. Unfortunately, this analysis could not be made with the data from the HHPD project [28], which served as data resource for the current study, as it was not designed to evaluate test-retest reliability.

As previously described, the MOH was developed in Germany; it was thereafter translated to English [11] and Swedish [6]. We used the Swedish version for the data collection whereas the English version was used when presenting the results in this paper. In general, translations of instruments to different languages are challenging and might influence the psychometric properties of the instrument [53]. A multi-lingual research team translated the MOH instrument from English to Swedish, but these translations could affect the psychometric properties of the MOH, which is important to keep in mind when interpreting our results.

Considering factors that could have influenced the results presented in this paper, one should be aware of that depressive symptoms are common among people with PD and is known to affect data collected with self-rated instrument. However, the prevalence of depression symptoms in this sample (see Table 1) was below the cut off value for the indication of depression (5 points) among people with PD [54] and should not affect the results substantially.

## Conclusions

The results of this study suggest that the 23-item version of the MOH is reliable and valid for use among people with PD. The results reveal a new clear three-dimensional structure with sub-scales that validly capture meaning of home in this population. However, as psychometric evaluation is a constant process, additional studies are needed to evaluate test-retest reliability and further confirm the overall psychometric quality of the 23-item MOH for use among people with PD. Still, this psychometric study paves the way for further research on the dynamics of housing and health in the PD population. The MOH instrument is foremost useful in research to promote the understanding of housing and health dynamics in different sub-groups of the ageing population, for example, people with PD. Knowledge emanating from such research has implications beyond clinical treatments and may serve to strengthen evidence to inform housing interventions and future housing policies supporting active and healthy ageing for this population.

## Acknowledgments

We thank the participants of the HHPD project and PhD statistician S. Ullén for valuable analysis guidance and interpretation of the results. This study was conducted within the context of the Centre for Ageing and Supportive Environments (CASE) and the Strategic Research Area in neuroscience (MultiPark) at Lund University, Sweden. The first author's learning process was supported by the National Graduate School on Ageing and Health (SWEAH).

## Author Contributions

**Conceptualization:** Nilla Andersson, Maria H. Nilsson, Björn Slaug, Frank Oswald, Susanne Iwarsson.

**Data curation:** Björn Slaug.

**Formal analysis:** Nilla Andersson, Björn Slaug.

**Funding acquisition:** Maria H. Nilsson, Susanne Iwarsson.

**Methodology:** Nilla Andersson, Maria H. Nilsson, Björn Slaug, Frank Oswald, Susanne Iwarsson.

**Project administration:** Maria H. Nilsson, Susanne Iwarsson.

**Software:** Björn Slaug.

**Supervision:** Maria H. Nilsson, Björn Slaug, Susanne Iwarsson.

**Validation:** Nilla Andersson, Maria H. Nilsson, Björn Slaug, Frank Oswald, Susanne Iwarsson.

**Visualization:** Nilla Andersson, Björn Slaug.

**Writing – original draft:** Nilla Andersson.

**Writing – review & editing:** Maria H. Nilsson, Björn Slaug, Frank Oswald, Susanne Iwarsson.

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
