## [Decision Letter · Decision Letter 0]

5 Jun 2020

PONE-D-20-10176

The Meaning of Home Questionnaire revisited: Psychometric analyses among people with Parkinson´s Disease reveals new dimensions

PLOS ONE

Dear Dr. Andersson,

Thank you for submitting your manuscript to PLOS ONE. After careful consideration, we feel that it has merit but does not fully meet PLOS ONE’s publication criteria as it currently stands. Therefore, we invite you to submit a revised version of the manuscript that addresses the points raised during the review process.

We look forward to receiving your revised manuscript.

Kind regards,

Stefan Hoefer

Academic Editor

PLOS ONE

Journal Requirements:

Reviewers' comments:

Reviewer's Responses to Questions

**Comments to the Author**

1. Is the manuscript technically sound, and do the data support the conclusions?

Reviewer #1: Partly

Reviewer #2: Yes

2. Has the statistical analysis been performed appropriately and rigorously? 

Reviewer #1: Yes

Reviewer #2: Yes

3. Have the authors made all data underlying the findings in their manuscript fully available?

Reviewer #1: No

Reviewer #2: No

4. Is the manuscript presented in an intelligible fashion and written in standard English?

Reviewer #1: Yes

Reviewer #2: Yes

5. Review Comments to the Author

Reviewer #1: In the present paper, the authors investigated the psychometric properties of the “Meaning of Home” (MOH-)Questionnaire, using a medium sized study sample of inpatients with Parkinson’s Disease (PD) from Sweden. To do so, the authors conducted several analyses including factor analyses as well as the assessment of construct validity and analyses of correlations, testing for predefined hypotheses. As main findings, they present a new factorial structure of the MOH in this population, and they report significant albeit low to moderate correlations of the MOH with the external measures chosen.

The manuscript is well written and also well structured. It covers a relevant scientific topic and its methodology is largely sound. However, I have some concerns which I would like to share with the authors below.

Major Comments:

1. My main concern is the possible mixing of similar but nevertheless different concepts, since the authors, despite all methodological care, do not present any measure of discriminant validity besides convergent validity. However, this is essential, since in order to assess the applicability of an instrument it is also necessary to estimate the ability of the scale to distinguish the concept under study from other (similar) concepts. Why did the authors decide to refrain from such a procedure?

2. The most prominent example is QoL, which has been used by the authors as a criterion for convergent validity, and the authors also note that there are great similarities in content between the two concepts. In fact, one could even provocatively ask whether the two are not the same and, conversely, whether one still needs to assess housing perceptions if the quality of life of PD patients is already being assessed anyway (which is now standard practice in both clinical care and clinical trials).

3. Another important point in this context is depression, which occurs very frequently in PD and is known to massively influence a large number of perception processes. Apparently, however, depression was neither assessed in the participants nor is this fact critically evaluated by the authors.

4. Overall, in light of the well written introduction and method section (with minor restrictions as listed below) the discussion is not yet fully developed and needs to be revised and sharpened. For example, I find the discussion of the factorial structure of the MOH as found by the authors excellent. On the other hand, I would have liked to see a much more critical discussion by the authors about the significant but nevertheless very low correlations in the results. Here, a clearer evaluation of these results with regard to their practical implications would have been very valuable.

Minor comments:

5. The introduction sections seems to be a little too lenghty to me and could be more streamlined a little bit. This applies above all to the presentation of the development history of the MOH, which is very detailed. At the same time, this is at the expense of a description of the practical and/or socioeconomic implications of assessing housing perceptions, which could be presented more clearly in return.

6. The drop-out rate as described in the methodology seems relatively high (though not necessarily exceptional). Now it is in the nature of drop-outs that there is little data available about them. However, can it still be ruled out anyway that there was a systematic loss of participants (e.g. in terms of age, gender, etc.)?

7. Obviously, there is a typo on page 17, line 13 (“…. could leading to negative...”)

8. The interim conclusion which is presented on page 18 (lines 4-7) and which states that the removal of five items makes it possible for more frailed participants, seems a little bit exaggerated to me. From my clinical experience with the population under study, shortening an instrument by less than 20% does not have such an impact.

Reviewer #2: 02-June-2020

PONE-D-20-10176

TITLE: “The Meaning of Home Questionnaire revisited: Psychometric analyses among people with Parkinson´s Disease reveals new dimensions”

Reviewer:

Comments to the Author

The authors aimed to create a psychometrically sound instrument to study perceptions of home in people with Parkinson’s disease (PD). This approach in environmental gerontology is novel and instruments for the assessment of perceptions of home are lacking. The authors collected data from a sample of 245 PD patients about perceptions of home captured by the Meaning of Home Questionnaire (MOH). Subsequently, they did a psychometric analysis of internal consistency, item-scale correlations and exploratory factor analysis. The found a new three-factor solution of MOH and satisfactory internal consistency level for the whole questionnaire. Overall, a solid piece of psychometric work resulting in the creation of a new instrument (MOH) for the assessment of perceptions of home in PD but possibly also in other clinical populations. However, I found some possible amendments or omissions and would beg the authors to improve or correct them. Therefore, I recommend a major revision.

MAJOR COMMENTS

Introduction

1. Could you please be more specific regarding the question of “perceived housing” and its clinical utility in PD or other diseases? What is supposedly the clinical utility of measurement of “perceived housing”?

2. Could you also be more specific in the formulation of “knowledge gap” regarding “perceived housing” in general and in PD in particular?

Materials and Methods

3. P. 8, l. 6: “The MOH is interview administered… “ I am sorry, I do not understand if MOH is a self-report scale or if it is an interview done by an expert on scaling with the patient (not a self-report)?

4. Second, who is allowed (what kind of professional) do the interview?

5. Third, the part of the sentence “The MOH is interview administered… ” does not seem to be grammatically correct.

6. Regarding Statistical analyses and reliability (internal consistency) in particular, I would welcome also other internal consistency measures (beyond Cronbach’s alpha) in accordance with EFPA guidelines for psychological and educational tests (Version 4.2.6; http://www.efpa.eu/professional-development/assessment), suggested by the Board of Assessments of the European Federation of Psychologists’ Associations (EFPA, 2013).

7. Can you add also L-Dopa equivalent and other medication in your descriptive statistics of PD sample?

Discussion

In Discussion (also in the analysis of results), I am missing besides the interpretation of the psychometric analysis some discussion of relation (e.g., a simple correlation analysis) between clinical measures of PD (L-Dopa equivalent, H-Y staging, PD duration) and MOH.

8. Why was MOH tested in PD sample?

9. And is MOH related to some clinical aspects of the disease?

10. What is the difference between perception of home and ADL/IADL measures? These, in my opinion, may also include some aspects of perception of home. I mean, why is the perception of home an independent construct and should be measured independently from ADL and everyday functioning measures?

MINOR COMMENTS

None.

Tables: No comments.

Figures: A higher dpi would make the inspection easier, the figure is blurred.

6. PLOS authors have the option to publish the peer review history of their article (what does this mean?). If published, this will include your full peer review and any attached files.

Reviewer #1: No

Reviewer #2: No

---

## [Author Response · Author response to Decision Letter 0]

7 Sep 2020

Comments from editor:

Response: Thank you for the notification, the manuscript is updated in line with the style requirements.

Response: We have revised the text under data availability as follows:

“The data used in this study contains sensitive information about the study participants and they did not provide consent for public data sharing. The current approval by the Regional Ethical Review Board in Lund, Sweden (No. 2012/558) does not include data sharing. A minimal data set could be shared by request from a qualified academic investigator for the sole purpose of replicating the present study, provided the data transfer is in agreement with EU legislation on the general data protection regulation and approval by the Swedish Ethical Review Authority. 

Contact information:

Department of Health Sciences, Lund University

Box 157, 221 00 Lund, Sweden

Att. Christina Brogårdh, Head of Department: christina.brogardh@med.lu.se

Principal investigator: Maria_H.Nilsson@med.lu.se

Swedish Ethical Review Authority, Box 2110, 75 002 Uppsala, Sweden. 

Phone: +46 10 475 08 00”. 

Clarifying this further, the ethical application covering the current study was approved by the Regional Ethical Review Board in Lund (Sweden), but the permit granted does not cover data sharing. We are thereby not able to publicly share data from the project, although it can be requested from a qualified academic investigator and shared anonymized for sole purposes of replication of the procedures and results presented in this paper. The data contains sensitive information about the participants, such as health variables (e.g. functional status, cognitive status).

Comments of reviewer 1

Major Comments:

1. My main concern is the possible mixing of similar but nevertheless different concepts, since the authors, despite all methodological care, do not present any measure of discriminant validity besides convergent validity. However, this is essential, since in order to assess the applicability of an instrument it is also necessary to estimate the ability of the scale to distinguish the concept under study from other (similar) concepts. Why did the authors decide to refrain from such a procedure?

Response: Thank you for this comment. Even though studying discriminant validity was not part of our original plan for the current study, we have now added such analyses and results. The results clearly support the discriminant validity of the MOH. See adjustments in the aim statement, method, result and discussion sections, and in Table 4. 

2. The most prominent example is QoL, which has been used by the authors as a criterion for convergent validity, and the authors also note that there are great similarities in content between the two concepts. In fact, one could even provocatively ask whether the two are not the same and, conversely, whether one still needs to assess housing perceptions if the quality of life of PD patients is already being assessed anyway (which is now standard practice in both clinical care and clinical trials).

Response: If we interpret this comment correctly, it seems as there was a misunderstanding about Quality of Life, as we did not address this concept in our study. We did study life satisfaction [1], which is another concept. We argued in the manuscript that meaning of home is crucial for higher life satisfaction in old age [2], but this does not mean that these concepts are the same. Whereas meaning of home is related to the domain of housing, life satisfaction is related to life in general. This is supported by the results concerning convergent validity, showing that MOH and life satisfaction are related but clearly distinguished, with correlations coefficients ranging between: 0.31-0.43; for more information see Table 4. A medium amount of correlation means that there is some overlap, as housing is part of life, but nevertheless the two concepts are far from identical. 

3. Another important point in this context is depression, which occurs very frequently in PD and is known to massively influence a large number of perception processes. Apparently, however, depression was neither assessed in the participants nor is this fact critically evaluated by the authors.

Response: Thank you for the suggestion to consider depression in our study. We do not have data on depression diagnoses, but on depressive symptoms. Please, see Table 1 for descriptive data. Furthermore, we added a sentence about this in the Discussion, as follows:

“Considering factors that could have influenced the results presented in this paper, one should be aware of that depressive symptoms are common among people with PD and is known to affect data collected with self-rated instrument. However, the prevalence of depression symptoms in this sample (see Table 1) was below the cut off value for the indication of depression (5 points) among people with PD and should not affect the results substantially.” 

4. Overall, in light of the well written introduction and method section (with minor restrictions as listed below) the discussion is not yet fully developed and needs to be revised and sharpened. For example, I find the discussion of the factorial structure of the MOH as found by the authors excellent. On the other hand, I would have liked to see a much more critical discussion by the authors about the significant but nevertheless very low correlations in the results. Here, a clearer evaluation of these results with regard to their practical implications would have been very valuable.

Response: We have followed the suggestion to incorporate a more critical discussion on the correlations, with the two following sections in the Discussion:

“Regarding the validity evaluation of the 23-item MOH, most correlations with the variables chosen for these analyses supported the construct validity of this version of the instrument. The results showing that the strength of the correlations measuring construct validity is low [46] indicate that meaning of home is distinctly different from usability of the home, ADL dependence and life satisfaction, although related (see Table 4). Accordingly, these results are in line with the pre-defined hypotheses. Comparing with earlier studies in general populations samples of very old people, the correlations in the present study are on approximately similar levels [4,6]. Reflecting further on, for example, the relation of meaning of home with the well-established and clinically relevant indicators for ADL dependence, meaning of home is based on perceptions of meaningfulness related to the home [11, 12], emerging form psychological theory. In contrast, ADL is a largely physical aspect of performance in everyday activities [35, 36]. Thus, the two concepts have fundamentally different origins, although overlapping in real life situations, and – as demonstrated by previous as well as present results - measuring them requires different instruments.”

And:

“As to the fact that our hypothesis was not supported regarding two of the MOH sub-scales, it seems as other factors than time lived in the dwelling are related to perceiving one’s home as a prison or castle, respectively. One example of a redefined hypothesis is that for people living and ageing with a chronic, progressive disease, functional decline is related to perceiving the home as a prison. Obviously, more research is warranted to shed further light on those associations. 

As to discriminant validity, the very low and non-significant correlations between MOH sub-scale scores and number of environmental barriers demonstrate that even if the variables represent aspects of housing, self-perceived meaning of home is distinctly different from an objectively measurable variable capturing physical housing features.”

Furthermore, considering the suggestion of reasoning about the clinical implications, emanating from environmental gerontology the MOH instrument is mainly useful for research aiming to increase the understanding of housing and health dynamics. In order to accomplish such a goal, there is a need to learn more about a phenomenon such as meaning of home. Accordingly, the results of our study are predominantly useful in a research context evaluating the instrument to learn more about meaning of home in the PD population. We have clarified this in the Introduction (pages 3 and 5), Discussion (page 20) and Conclusion (page 22), please, see the yellow marked text. An example of this is the text in the Conclusion, with the following text:

“The MOH instrument is foremost useful in research to promote the understanding of housing and health dynamics in different sub-groups of the ageing population, for example, people with PD. Knowledge emanating from such research has implications beyond clinical treatments and may serve to strengthen evidence to inform housing interventions and future housing policies supporting active and healthy ageing for this population.”

Minor comments:

5. The introduction sections seems to be a little too lenghty to me and could be more streamlined a little bit. This applies above all to the presentation of the development history of the MOH, which is very detailed. At the same time, this is at the expense of a description of the practical and/or socioeconomic implications of assessing housing perceptions, which could be presented more clearly in return.

Response: We agree that the Introduction was a bit long, therefore we have shortened this section on pages 3, 4 and 5 (see the yellow markings). Although, the history of the instrument development is described in detail, it has not been summarized in any previous publication. Because meaning of home is not a well-known concept, we find this section useful for the reader to fully understand the nature and structure of the instrument.

Further we have added text on why it is important to know more about meaning of home for a PD population in the Introduction on pages 3 and 5 (see the yellow markings). 

6. The drop-out rate as described in the methodology seems relatively high (though not necessarily exceptional). Now it is in the nature of drop-outs that there is little data available about them. However, can it still be ruled out anyway that there was a systematic loss of participants (e.g. in terms of age, gender, etc.)?

Response: We think that the final sample in our study is representative for the PD population as it is close to what is found on the population level regarding age and sex [3]. Furthermore, we consider it as a strength that we included participants from all HY stages (I-V) representing the full disease severity spectrum and with a broad age range (45-93 years).

7. Obviously, there is a typo on page 17, line 13 (“…. could leading to negative...”)

Response: Thank you for your accuracy, corrected the typo.

“…could lead to negative…..”

8. The interim conclusion which is presented on page 18 (lines 4-7) and which states that the removal of five items makes it possible for more frailed participants, seems a little bit exaggerated to me. From my clinical experience with the population under study, shortening an instrument by less than 20% does not have such an impact.

Response: In large research projects the data collection arsenal is often massive, with many different assessments and questions. Accordingly, short and effective instruments are important to minimize the participant burden and overall need of resources. Especially in a vulnerable population such as people with PD this is important. Expressing that the suggested shortening of the MOH instrument “might make it possible” is a cautious conclusion, which we think is reasonable. 

Comments from reviewer 2

MAJOR COMMENTS

Introduction

1. Could you please be more specific regarding the question of “perceived housing” and its clinical utility in PD or other diseases? What is supposedly the clinical utility of measurement of “perceived housing”?

Response: As we have answered reviewer 1 (remark 4) regarding the clinical utility, the MOH instrument is not intended for clinical use but primarily for research aiming to get a deeper understanding about housing and health dynamics along the process of ageing. In such research, the phenomenon of meaning of home is a core concept (see e.g. Oswald et al., 2006) [4]. In the revised version of our manuscript, this has been clarified in the Introduction (pages 3 and 5), Discussion (page 20) and Conclusion (page 22), see the yellow marked text.

2. Could you also be more specific in the formulation of “knowledge gap” regarding “perceived housing” in general and in PD in particular?

Response: Regarding the knowledge gap of perceived housing in PD, we have evolved the text (see the yellow marked text) in the Introduction about this (the same text pieces to answer reviewer 1, remark 5).

Materials and Methods

3. P. 8, l. 6: “The MOH is interview administered… “ I am sorry, I do not understand if MOH is a self-report scale or if it is an interview done by an expert on scaling with the patient (not a self-report)?

Response: The MOH questionnaire was assessed face-to-face by project administrators asking questions to the participants during home visits. We have clarified that in the text with the following sentence:

“In this study, MOH questionnaire was assessed in face-to-face sessions during home visits”

4. Second, who is allowed (what kind of professional) do the interview?

Response: We have added a part of a sentence to answer this question, as follows: 

“The MOH is a questionnaire administered by specifically trained project assistants from different professional backgrounds.”

5. Third, the part of the sentence “The MOH is interview administered… ” does not seem to be grammatically correct.

Response: The sentence has been revised. Please, see the response to remark 3 (Reviewer 2).

6. Regarding Statistical analyses and reliability (internal consistency) in particular, I would welcome also other internal consistency measures (beyond Cronbach’s alpha) in accordance with EFPA guidelines for psychological and educational tests (Version 4.2.6; http://www.efpa.eu/professional-development/assessment), suggested by the Board of Assessments of the European Federation of Psychologists’ Associations (EFPA, 2013).

Response: We are aware of that several guidelines such as the one suggested by this reviewer do exist, for example, regarding internal consistency measures. For this study we used the COSMIN checklist (information now added in the Methods section, p. 8), which recommends using Cronbach’s alpha to evaluate internal consistency. As this is a well-known standard test, we do not think adding another internal consistency measure would improve the quality of the study. The added sentence in the Method sections, is as follows:

“The choice of statistical tests to evaluate the psychometric properties of the MOH instrument was based on the COSMIN checklist [40].”

7. Can you add also L-Dopa equivalent and other medication in your descriptive statistics of PD sample?

Response: With the overarching aim of the HHPD to generate knowledge on housing and health in people with PD with the explicit attention to PD specific symptomatology [5], we do not have access to data regarding medications in our dataset. As our aim was to evaluate the psychometric properties of an instrument addressing perceived aspects of housing among people with PD, information about medication is less relevant than in traditional PD research. Please note that the data on disease severity and disease duration is presented in Table 1. 

Discussion

In the Discussion (also in the analysis of results), I am missing besides the interpretation of the psychometric analysis some discussion of relation (e.g., a simple correlation analysis) between clinical measures of PD (L-Dopa equivalent, H-Y staging, PD duration) and MOH.

Response: We seriously considered this suggestion, but concluded that such analyses are beyond the scope and aim of the current study.

8. Why was MOH tested in PD sample?

Response: See answer on remark 2 (Reviewer 2).

9. And is MOH related to some clinical aspects of the disease?

Response: Considering this question, we would like to once again emphasise that targeting perceptions of meaning of home, at least at present, is a research interest rather than a clinical one. Still, as we chose the set of concepts used to evaluate the construct validity of the MOH instrument, we included ADL, that is, a variable representing an important clinical aspect of the disease. Accordingly, we now reflect upon this in the Discussion section, with the following text:

“Reflecting further on, for example, the relation of meaning of home with well-established and clinically relevant indicators for ADL dependence, meaning of home is based on perceptions of meaningfulness related to the home [11, 12], emerging form psychological theory. In contrast, ADL is a largely physical aspect of performance in everyday activities [35, 36]. Thus, the two concepts have fundamentally different origins, although overlapping in real life situations, and – as demonstrated by previous as well as present results – measuring them requires different instruments.”

10. What is the difference between perception of home and ADL/IADL measures? These, in my opinion, may also include some aspects of perception of home. I mean, why is the perception of home an independent construct and should be measured independently from ADL and everyday functioning measures?

Response: Please see our previous response to this (Remark 9, Reviewer 2). 

Even though the correlations between meaning of home and independence in ADL presented in our study were statistically significant they are low (rs = -0.18 to -0.33) and indicate that the concepts are different. We have incorporated a section about this in the Discussion.

MINOR COMMENTS

None.

Tables: No comments.

Figures: A higher dpi would make the inspection easier, the figure is blurred.

Response: This is taken care of and a high quality figure is included in the revised version. See figure 1.

References:

1. Fugl-Meyer AR, Melin R, Fugl-Meyer KS. Life satisfaction in 18- to 64-year-old Swedes: in relation to gender, age, partner and immigrant status. J Rehabil Med. 2002;34(5):239-46.

2. Oswald F, Wahl HW. Housing and health in later life. Rev Environ Health. 2004;19(3-4):223-52.

3. Van Den Eeden SK, Tanner CM, Bernstein AL, Fross RD, Leimpeter A, Bloch DA, et al. Incidence of Parkinson’s Disease: Variation by Age, Gender, and Race/Ethnicity. American Journal of Epidemiology. 2003;157(11):1015-22.

4. Oswald F, Schilling O, Wahl H-W, Fänge A, Sixsmith J, Iwarsson S. Homeward bound: Introducing a four-domain model of perceived housing in very old age. Journal of Environmental Psychology. 2006;26(3):187-201.

5. Nilsson MH, Iwarsson S. Home and health in people ageing with Parkinson's disease: study protocol for a prospective longitudinal cohort survey study. BMC Neurology. 2013;13(142):1-9.

---

## [Decision Letter · Decision Letter 1]

22 Oct 2020

PONE-D-20-10176R1

The Meaning of Home questionnaire revisited: Psychometric analyses among people with Parkinson´s Disease reveals new dimensions

PLOS ONE

Dear Dr. Andersson,

Thank you for submitting your manuscript to PLOS ONE. After careful consideration, we feel that it has merit but does not fully meet PLOS ONE’s publication criteria as it currently stands. Therefore, we invite you to submit a revised version of the manuscript that addresses the points raised during the review process.

We look forward to receiving your revised manuscript.

Kind regards,

Stefan Hoefer

Academic Editor

PLOS ONE

Additional Editor Comments (if provided):

Thank you very much for addressing the points of both reviewers. I kindly ask you to take up the minor remark of reviewer 2, whether the instrument is in the public domain or not.

Reviewers' comments:

Reviewer's Responses to Questions

**Comments to the Author**

1. If the authors have adequately addressed your comments raised in a previous round of review and you feel that this manuscript is now acceptable for publication, you may indicate that here to bypass the “Comments to the Author” section, enter your conflict of interest statement in the “Confidential to Editor” section, and submit your "Accept" recommendation.

Reviewer #1: All comments have been addressed

Reviewer #2: All comments have been addressed

2. Is the manuscript technically sound, and do the data support the conclusions?

Reviewer #1: Yes

Reviewer #2: Yes

3. Has the statistical analysis been performed appropriately and rigorously? 

Reviewer #1: Yes

Reviewer #2: Yes

4. Have the authors made all data underlying the findings in their manuscript fully available?

Reviewer #1: No

Reviewer #2: Yes

5. Is the manuscript presented in an intelligible fashion and written in standard English?

Reviewer #1: Yes

Reviewer #2: Yes

6. Review Comments to the Author

Reviewer #1: (No Response)

Reviewer #2: I have no further comments. The authors replied satisfactorily to my review.

MINOR COMMENTS

Thank you for clarifying the difference between IADL and perceptions of housing and other questions. Maybe, also a note if the MOH is in public domain or if the instrument is copyrighted would be helpful.

7. PLOS authors have the option to publish the peer review history of their article (what does this mean?). If published, this will include your full peer review and any attached files.

Reviewer #1: No

Reviewer #2: No

---

## [Author Response · Author response to Decision Letter 1]

9 Nov 2020

2020-11-05

Dear Editor in Chief,

Please find enclosed the revised version of our manuscript with the title “The Meaning of Home Questionnaire revisited: Psychometric analyses among people with Parkinson´s Disease reveals new dimensions" (PONE-D-20-10176).

We thank the reviewer for the final comment for improvements of our paper. Below follow our point-by-point responses to the comments of the reviewer, and a description of the revisions in the manuscript. In the uploaded revised manuscript version, all changes made are highlighted in yellow. 

Alongside the revision based on the reviewers´ suggestions, we made minor text optimization throughout (however, not marked in the revised version of the manuscript) and uploaded our figure in PACE as recommended.

We hope that the revisions and responses will be sufficient for our manuscript to be accepted for publication in PLOS ONE. We are looking forward to further communication on this matter.

On behalf of myself and the co-authors,

Nilla Andersson

First author, PhD student

Comment from a reviewer:

Comment 6. Clarify if the MOH is in public domain or if the instrument is copyrighted.

Response: Thank you for this suggestion; we have incorporated information on this in the manuscript under the method section, as follows:

“Psychometric properties of the MOH questionnaire for use in the general population of older people have been reported earlier [6]. The questionnaire [32] is available in English, Swedish, and German versions for research application (on request only, to author F.O.).”

---

## [Editor Report · Decision Letter 2]

10 Nov 2020

The Meaning of Home questionnaire revisited: Psychometric analyses among people with Parkinson´s Disease reveals new dimensions

PONE-D-20-10176R2

Dear Dr. Andersson,

We’re pleased to inform you that your manuscript has been judged scientifically suitable for publication and will be formally accepted for publication once it meets all outstanding technical requirements.

Kind regards,

Stefan Hoefer

Academic Editor

PLOS ONE
---

## [Editor Report · Acceptance letter]

25 Nov 2020

PONE-D-20-10176R2 

The Meaning of Home questionnaire revisited: Psychometric analyses among people with Parkinson´s Disease reveals new dimensions 

Dear Dr. Andersson:

I'm pleased to inform you that your manuscript has been deemed suitable for publication in PLOS ONE. Congratulations! Your manuscript is now with our production department. 

Kind regards, 

on behalf of

Dr. Stefan Hoefer 

Academic Editor

PLOS ONE